# Laboratory Experimental Study on Polymer Flooding in High-Temperature and High-Salinity Heavy Oil Reservoir

**Fujian Zhang [1,2,3], Youwei Jiang [2,3], Pengcheng Liu [1,*], Bojun Wang [2,3], Shuaishuai Sun [1], Daode Hua [1] and Jiu Zhao [4]**

1 School of Energy Resources, China University of Geosciences, Beijing 100083, China
2 Research Institute of Petroleum Exploration and Development, PetroChina, Beijing 100083, China
3 State Key Laboratory of Enhanced Oil Recovery, PetroChina, Beijing 100083, China
4 Beijing Techvista Scientific Co., Ltd., Beijing 100083, China
* Correspondence: lpc@cugb.edu.cn

**Abstract:** Polyacrylamide (HPAM) and other traditional polymers have poor temperature resistance and salinity tolerance and do not meet the needs of high-temperature and high-salinity reservoirs. In this study, a new temperature-resistant and salinity-tolerant polymer QJ75-39 was synthesized using acrylamide (AM) as a hydrophilic monomer, 1-acrylamide-2-methylpropanesulfonic acid (AMPS) and N-vinylpyrrolidone (NVP) as functional monomers and DS-16 as a hydrophobic monomer. Through laboratory experiments, the properties (temperature resistance, salinity tolerance and aging stability), polymer injection and core displacement effect of the polymer were studied. The experimental results showed that the new polymer could meet the needs of polymer flooding technology in high-temperature and high-salinity reservoirs. Experiments showed that the polymer had a temperature resistance of 95 °C and a salinity tolerance of $1.66 \times 10^5$ mg/L. When the temperature was 95 °C and the TDS was 55,376.8 mg/L, the viscosity of the polymer was 31.3 mPa s, and the viscosity remained above 30 mPa·s after aging for 60 days. The polymer had good injectivity between 300 and 600 mD, and the injection pressure could reach equilibrium quickly. The oil recovery effectively increased with the grsowth in the amount of injected polymer. When the injection amount was 0.5 PV, the enhanced oil recovery was 20.65%. This study is of great significance for the application and popularization of polymer flooding technology in high-temperature and high-salinity reservoirs.

**Keywords:** high temperature and salinity; heavy oil; polymer flooding; enhanced oil recovery





## 1. Introduction

Heavy oil resources are abundant and widely distributed in the United States, Canada, Venezuela, Russia and China. Due to the high viscosity of heavy oil, water flooding is not a good way to develop heavy oil resources. The main methods include steam flooding, steam huff and puff and steam-assisted gravity drainage (SAGD). Many researchers have shown that thermal recovery is not advisable in deep heavy oil reservoirs. This is due to serious heat loss from formation and the wellbore, the low dryness of steam injection, the limited heating range and the poor development effect [1–4]. For deep heavy oil reservoirs, other methods need to be considered [5–7].

Polymer flooding is one chemical EOR method used to improve macroscopic oil sweep efficiency and hinder the fingering phenomena that occur due to highly permeable regions and the water cut [8–10]. At present, polymer flooding has achieved good development results in many heavy oil blocks, such as Pelican Lake [11,12], Mooney, Seal [13], Marmul [14–16], Bohai [17] and Medicine Hat [18]. The viscosity of the underground crude oil in the experimental area of the Pelican Lake oilfield is 1200~1800 mPa s. Since polymer flooding was implemented in 2005, the oil recovery in the experimental block has increased by 20% [11,12]. In 1956, Marmul oilfield was discovered in the south of Oman. The viscosity of crude oil is 80~110 mPa·s, and polymer flooding has achieved good development

results. The oil recovery following polymer flooding is 46% [14–16]. The average viscosity of crude oil in Bohai oilfield is 70 mPa·s. After the success of a test, polymer flooding was carried out with four injection and six production methods in 2005. The water cut decreased by 10% after polymer flooding, and the crude oil production per well increased by $1.77 \times 10^4$ m$^3$ [17]. Thus, it can be seen that polymer flooding can effectively enhance oil recovery in heavy oil reservoirs.

Hydrolyzed polyacrylamide (HPAM) has been used for the most oilfield projects because of its low cost and great viscoelasticity. It can increase the viscosity of the displacing phase while reducing the water/oil mobility ratio so as to enlarge the swept volume [19–21]. However, the hydrolysis reaction of HPAM is very rapid, and its stability is poor in high-temperature and high-salinity reservoirs. The carboxyl group produced by hydrolysis is easy to precipitate with large amounts of $Ca^{2+}$ and $Mg^{2+}$, which destroys the formation conditions and reduces the temperature resistance and salinity tolerance of the polymer; thus, it cannot meet the requirements of high-temperature and high-salinity reservoirs [22–27].

In order to improve the properties of HPAM, much research has been undertaken on changing the chemical structure of HPAM in the process of polymer flooding [24–28]. Researchers have mainly focused on replacing various AM moieties in the main chain. The properties of the copolymers can be optimized by adding different functional groups to HPAM to obtain better stability. Polymers containing sulfonate groups are expected to offer higher stability in solution and are tolerant to high salinity [29].

In 1986, Evani first proposed the concept of hydrophobically associating water-solute polymer (HAWPs), and they were successfully synthesized through the association of hydrophobic groups in aqueous solution. HAWP is a functional polymer polymerized by two types of monomers, hydrophilic and hydrophobic, under the action of the initiator and the temperature [30]. Due to the hydrophobic groups, there are intramolecular and intermolecular associations in the aqueous solution system. At a certain polymer concentration, similarly to the micelles formed by surfactants above their critical micelle concentration, the hydrophobic groups in these polymers aggregate through intermolecular hydrophobic interaction and tend to associate in aqueous solution [31]. This leads to the formation of multi-molecular associations, which can greatly enhance the stability. Due to their stability in salinity compared to unmodified polymers, in addition to their enhanced viscosity, HAWPs have great potential for use in high-temperature and high-salinity reservoirs [32].

At present, HAWPs also have several problems, including:

(1) Poor solubility. With the increase in the polymer molecular weight and hydrophobic group content, the solubility becomes lower and lower. Some researchers enhance solubility by adding surfactants;

(2) Poor injectivity. Although the resistance coefficient (RF) and residual resistance coefficient (RRF) of hydrophobically associated polymers are improved during displacement, their injectivity is not as good as that of HPAMs [33];

(3) The temperature and salt resistance of hydrophobically associating polymers can be further improved.

At present, the commonly used monomers are AMPS, NVP and sodium p-styrene sulfonate (SSS). In order to solve the above problems, we have carried out a series of studies [34,35].

Block A of Lukeqin oilfield (Xinjiang, China) is a typical deep heavy oil reservoir with high temperature (85~100 °C), high salinity (8.3~16.02 × 10$^4$ mg/L, CaCl$_2$) and deep burial depth (2300 m~3100 m). It has a thick pay zone of 20$^+$ m, an average porosity of 27.31%, a live oil viscosity of 526 mPa·s and a saturation pressure of 4.1 MPa. The reservoir heterogeneity is strong. With the deepening of the oilfield, the formation energy has rapidly decreased, production in Lu block A has decreased rapidly, the number of low-yield and low-efficiency wells has continued to increase, and the water cut (65%) of some wells has increased rapidly. SAGD technology is usually used for high-temperature and -salinity heavy oil reservoirs, but Lukeqin oilfield has a deep burial depth and extensive wellbore heat loss, making it not suitable for SAGD technology [36–38]. According to predictions,

the ultimate recovery from water flooding in Lu block A is less than 15%, so there is an urgent need for a new method to improve the development effect.

In this study, we conducted a series of experiments to analyze the feasibility of polymer flooding in Lukeqin oilfield. A new type of temperature-resistant and salinity-tolerant polymer was designed and synthesized, and its structure and properties were tested. The effects of temperature, salinity and aging stability on the polymer were evaluated, and its injectivity and oil displacement ability were also evaluated through laboratory experiments. This study has great significance for the application and popularization of polymer flooding technology in high-temperature and high-salinity reservoirs.

## 2. Experiments

### 2.1. Preparation and Characterization of Polymer

In light of the high-temperature and -salinity reservoir conditions in Lu block, development of a temperature-resistant and salinity-tolerant polymer development should meet the following requirements: (1) the intrinsic viscosity should be $\leq 2500$ mL/g; (2) the polymer should have good solubility and the dissolution time of the polymer should be $\leq 3$ h; (3) the polymer should have good viscosity enhancement, and the apparent viscosity of 2000 mg/L polymer solution should be $\geq 30$ mPa·s; (4) the polymer should have good aging stability, and the viscosity of 2000 mg/L solution should be $\geq 25$ mPa·s after 60 days of aging; (5) the oil recovery should increase by more than 10%. In order to meet the above conditions, the temperature and salinity tolerance of HPAM was improved from the point of view of molecular structure, and the static properties of the polymer were tested.

#### 2.1.1. Materials

The main materials in the experiment were acrylamide (AM, 99%), 1-acrylanmido-2-methylpropanesulfonic acid (AMPS, 99%), N-vinyl-pyrrolidone (NVP, 99%, Beijing Chemical Plant, Beijing, China) and HPAM ($1400 \times 10^4$, 99%) (Beijing Hengju Chemical Co., Ltd, Beijing, China).

The cosolvents were fatty alcohol polyoxyethylene ether (AEO) A1, A2, A3, B1, B2 and B3 (Beijing Chemical Plant).

The diallyl hydrophobic monomers were DS-10, DS-12, DS-14 and DS-16 (99%) (Southwest University of Petroleum).

Other chemical reagents used were $(NH_4)_2S_2O_8$ (99%), $K_2S_2O_8$ (99%), $NaHSO_4$ (99%), Di-tert-butyl peroxide (DTBT, 99%), NaCl (99%), $CaCl_2 \cdot 2H_2O$ (99%), $MgCl_2 \cdot 6H_2O$ (99%), $NaHCO_3$ (99%) and NaOH (99%) (Shenzhen Carbon Top Technology Co., Ltd, Shenzhen, China).

#### 2.1.2. Experimental Apparatus

The experimental apparatus included an electronic balance, LVDV-IIIU rotational viscometer (Brookfield Co., Ltd, Middleboro, MA, USA) and Tensor-37 Fourier-transform infrared spectrometer (Barker Co., Ltd, Foshan, China).

#### 2.1.3. Preparation of Polymer

(1) Figure 1 shows the design idea for the new polymer. Polyacrylamide is severely hydrolyzed in high-temperature and salinity reservoirs, and the more severe the hydrolysis, the easier it is for precipitation to occur from the solution. Furthermore, the solution viscosity retention is low. In order to obtain the desired temperature and salt resistance performance in the polyacrylamide, the polymer was designed according to the molecular design principle [39].

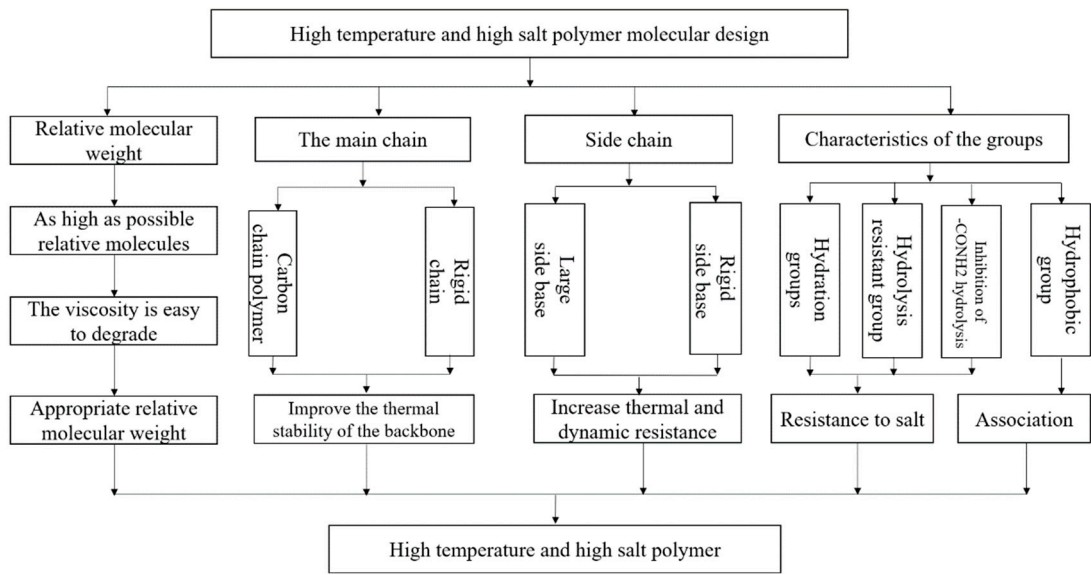

**Figure 1.** Diagram of the polymer design.

(i)   Matching of initiations and polymerization conditions.

The molecular weights of polymers obtained using different initiation systems and polymerization conditions are very different, so it was necessary to screen suitable initiations and polymerization conditions.

(ii)   Addition of temperature-resistance and salinity-tolerance monomers.

The stronger the rigidity of polymer main chains with the same molecular weight, the larger the hydrodynamic size and the higher the stability. AMPS can effectively improve rigidity. The sulfonic group in AMPS can effectively improve temperature resistance and salinity tolerance.

NVP can inhibit the hydrolysis of amides. The copolymerization of NVP and AM can obviously inhibit the hydrolysis of amides in polymer molecules and results in obvious salinity tolerance.

(iii)   Addition of hydrophobic monomers.

The hydrophobic association forms a reversible physical cross-linking network structure that improves the temperature resistance and salinity tolerance.

(2)   After determining the polymerization method and initiation system, various functional monomers were copolymerized using AM, and a new type of polymer was synthesized under suitable reaction conditions [40]. In the process of the experiment, each item had to be accurate. The preparation steps were as follows:

(i)   Quantitative AM, AMPS, a hydrophobic monomer and an appropriate amount of deionized water were added to the reaction bottle, which was stirred to dissolve the mixture;

(ii)   The pH value of the system was adjusted to the desired range with NaOH solution and the reaction temperature set for 40 °C;

(iii)   An appropriate amount of initiator (the initiator accounted for 0.03% of the total mass of the monomer) was added over 10 min through $N_2$ to remove dissolved $O_2$ from the water, and the temperature was recorded;

(iv)   The gel-like elastic target product was obtained by reacting at 40 °C for 6~8 h. The polymer samples were obtained by chopping, drying, grinding and sieving. Figure 2 shows the molecular structure diagram for the target polymer;

(v)   Using an appropriate number of samples, a Fourier infrared spectrometer was used to scan in the range of wave numbers 4000~400 cm$^{-1}$ (resolution 0.01 cm$^{-1}$) and the infrared spectrum of the sample was obtained.

**Figure 2.** Structural schematic diagram for the target polymer.

2.1.4. Test of Polymer

In this study, the properties of the polymer were tested. The measurement of the viscosity had to be repeated three times, and the average of three measurements was used. The temperature error was 0.1 °C. The specific schemes were as follows:

(1)  Viscosity–temperature characteristics: after the polymer solution was aged for 12 h, the viscosity was respectively measured at 40 °C, 60 °C, 80 °C and 95 °C (7.34 $s^{-1}$);

(2)  Salinity tolerance: a polymer solution with 2000 mg/L concentration was prepared with the different concentrations of simulated formation water, and their viscosities were measured at 95 °C;

(3)  Aging stability: a polymer solution was sealed in a bottle filled with nitrogen and placed in an incubator at 95 °C. On days 0, 3, 5, 7, 15, 30, 45 and 60, the viscosity was measured.

*2.2. Oil Displacement Performance Evaluation*

2.2.1. Materials

Oil sample: the viscosity of the underground crude oil from a heavy oil reservoir in Lu block A was 526 mPa·s. Table 1 shows the properties and composition of crude oil.

**Table 1.** Properties and composition of crude oil.

| Density | Live Oil Viscosity | Solidify Point | Wax Content | Asphaltene | Non-Hydrocarbon |
|---|---|---|---|---|---|
| $(g/cm^3)$ | (mPa·s) | (°C) | (%) | (%) | (%) |
| 0.9668 | 526 | 14 | 3.11 | 18.32 | 26.26 |

Water sample: to configure the formation simulation water, a water sample was taken with reference to the water quality analysis conditions of Lu block A; prepared with NaCl, $CaCl_2$ and $MgCl_2$ solution; filtered; and stored in an airtight container. The total salinity was 55,376 mg/L. Table 2 shows the ion composition of the water sample.

**Table 2.** Ion composition of water sample.

| Ion Species | $K^+$ | $Na^+$ | $Ca^{2+}$ | $Mg^{2+}$ | $Cl^-$ | $SO_4^{2-}$ | TDS |
|---|---|---|---|---|---|---|---|
| Content (mg/L) | 194.1 | 22,397.23 | 5964.1 | 275.4 | 26,538.76 | 7.21 | 55,376.8 |

Polymer solution: 2000 mg/L. Core: artificial core A—300 mD, diameter: 2.5 cm, length: 30 cm; artificial core B—600 mD, diameter: 2.5 cm, length: 30 cm (Beijing Huarui Xincheng Technology Co., Ltd, Beijing, China). Figure 3 shows the physical diagram of cores.

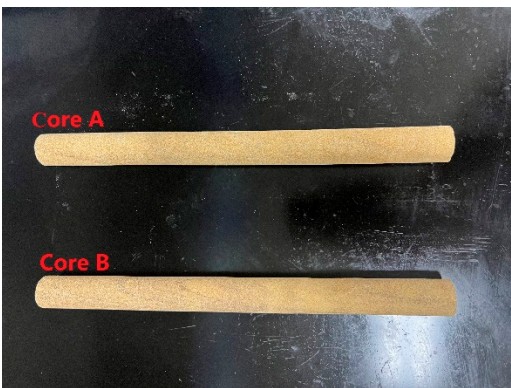

**Figure 3.** Physical picture of cores. Core A-300mD, Core B-600mD.

### 2.2.2. Experimental Apparatus

Figure 4 shows the flowchart of the physical model displacement device. It consisted of four parts:

(1) Solution injection system: the system consisted of three piston containers and an ISCO pump;
(2) Pipeline and reservoir simulation system: the system consisted of a core holder and an oven. The oven was used to keep the temperature constant during the experiment;
(3) Decomposition product collection and measurement system: the system consisted of a collection device, a gas flow meter, a backpressure valve, a pressure reducing valve, two pressure sensors and three pressure gauges;
(4) Pipeline cleaning system: the system consisted of an ISCO pump and a water tank.

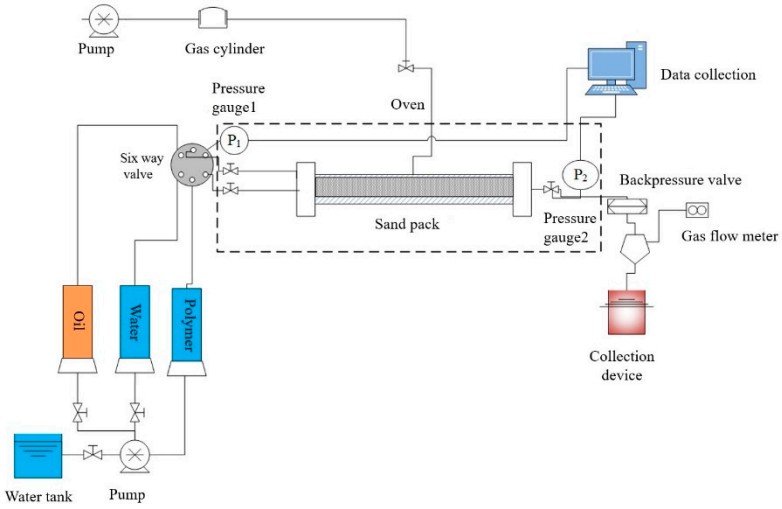

**Figure 4.** The flowchart of the physical model displacement device.

### 2.2.3. Polymer Injection Capability Experiment

The main procedures were as follows:

(1) The model was vacuumed and saturated with formation water at room temperature, and the model pore volume was obtained and water permeability measured;
(2) The formation water was injected under the conditions of constant temperature (95 °C) and constant speed (0.5 mL/min), and the stable injection pressure was recorded;
(3) The polymer solution was injected into the core at the same speed, and the stable polymer injection pressure was recorded;
(4) Water was re-injected into the core at the same speed, and the stable polymer injection pressure was recorded;

(5) The resistance coefficient *RF* and the residual resistance coefficient *RRF* were calculated:

$$RF = \frac{K_w}{\mu_w} / \frac{K_P}{\mu_p} \tag{1}$$

$$RRF = K_{wi} / K_{wa} \tag{2}$$

where $K_w$ and $K_p$ are the permeability of water and polymer, respectively; $\mu_w$ and $\mu_p$ are the viscosity of water and the viscosity of polymer, respectively; and $K_{wi}$ and $K_{wa}$ are the permeability of water before and after polymer flooding.

### 2.2.4. Core Displacement Experiment

Through the core displacement experiment, the oil displacement effect of the polymer was studied under ideal conditions. It is best to use a natural core in such experiments, but an artificial core can also be representative. The main procedures were as follows:

(1) The model was vacuumed and saturated with formation water to obtain the model pore volume, and the water permeability was measured at room temperature;
(2) The model was saturated with oil at 95 °C, and the oil saturation was calculated;
(3) Water flooding was carried out until a 98% water cut was reached at constant temperature (95 °C) and constant speed (0.5 mL/min);
(4) Polymer flooding with a given PV number (0.2 PV, 0.3 PV and 0.5 PV) was carried out at 0.5 mL/min and 95 °C and, subsequently, water flooding was carried out until the water cut reached more than 98%.

## 3. Results and Discussion

### 3.1. Analysis of the Polymer Preparation

### 3.1.1. The Effect of the Hydrophobic Monomer

Diallyl hydrophobic monomers have good water solubility, hydrolysis resistance and thermal stability. In order to improve the reaction activity, we used DS-n diallyl hydrophobic monomers. Figure 5 shows their molecular structure. We used the DS-10, DS-12, DS-14, DS-16 and DS-18 hydrophobic monomers. The results show that, with a longer hydrophobic chain, the polymer had a longer dissolution time. When the hydrophobic carbon chain was longer than 18 (Ds-18), the dissolution time was more than 4 h. Considering the dissolution factor, only DS-10, DS-12, DS-14 and DS-16 could be selected.

**Figure 5.** DS-n molecular structure.

Figure 6 shows the viscosities of four hydrophobic carbon chain monomers with different lengths at different concentrations.

As shown in Figure 6, the DS-n hydrophobic monomers are the same hydrophobic monomer but have different chain lengths. According to the experimental data, the viscosity of the polymer increased with the lengthening of the hydrophobic carbon chain. The viscosity of DS-16 was the highest, so DS-16 was chosen as the hydrophobic monomer for the target polymer.

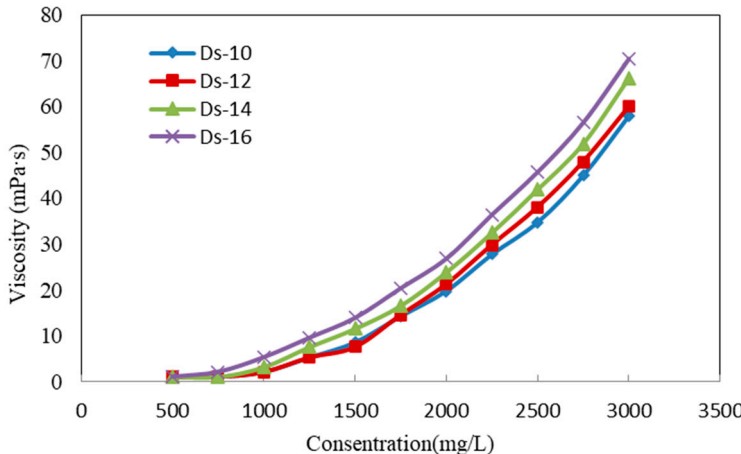

**Figure 6.** Effects of hydrophobic monomers with different carbon chain lengths on temperature and salinity tolerance of polymers. Carbon chain length: DS16 > DS14 > DS12 > DS10.

### 3.1.2. The Effect of the Initiator

Figure 7 shows the rise in temperature during the polymerization of different initiators.

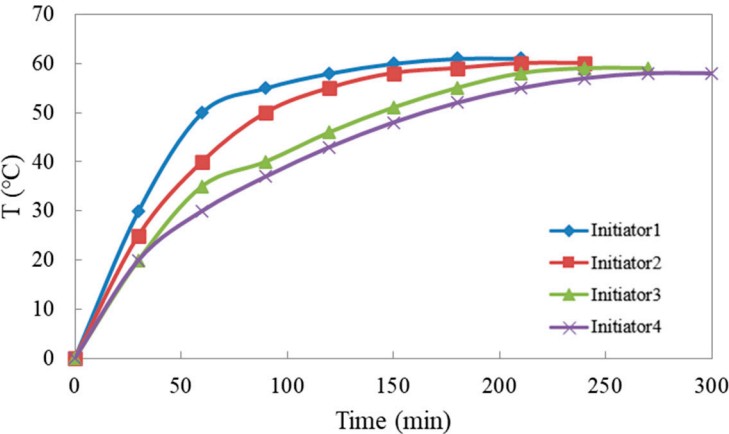

**Figure 7.** Temperature rise curve for initiator polymerization. Initiator1: $K_2S_2O_8$ + $NaHSO_4$; Initiator2: $(NH_4)_2S_2O_8$ + $NaHSO_4$; Initiator3: DTBP + $NaHSO_4$; Initiator4: $K_2S_2O_8$ + DTBP + $NaHSO_4$.

As shown in Figure 7, the reaction activities of $K_2S_2O_8$ + $NaHSO_4$ and $(NH_4)_2S_2O_8$ + $NaHSO_4$ were lower and the initiation temperatures higher, so it is not easy to increase the molecular weight of the polymer. However, the reaction activity of DTBT + $NaHSO_4$ was too high and a large number of free radicals were produced in a short time, resulting in too fast a polymerization rate and too low a molecular weight for the polymer. The insufficient reaction phenomenon was observed when the amount of the initiating system was reduced. Therefore, by selecting the $K_2S_2O_8$ + DTBP + $NaHSO_4$ composite initiation system, the problem of the poor solubility of the AIBN initiator was solved, the problems of rapid heating in the reaction process of DTBT + $NaHSO_4$ and the excessive initiation temperature of $K_2S_2O_8$ + $NaHSO_4$ were avoided and the molecular weight of the polymer was increased.

According to the experimental results, the $K_2S_2O_8$ + DTBP + $NaHSO_4$ system had the smallest temperature rise as the initiator.

### 3.1.3. The Effect of the Cosolvent on the Solubility of the Polymer

Solubility is an important index when evaluating the field application of polymers. The effect of cosolvents on the solubility of hydrophobically associating polymers results from the hydrophobic associating action of the polymer, and the hydrophobic associating

point is the hydrophobic micro-region, which is not easy to hydrate [41,42]. Therefore, the effect of association can be weakened by adding appropriate cosolvents in the process of hydrolytic drying.

Table 3 lists the effects of six kinds of cosolvents on the solubility of polymers. The addition of cosolvents is generally not conducive to increasing the molecular weight of polymers and, further, affects the viscosity of polymer solutions, but it is beneficial for improving the solubility of polymers. Non-ionic cosolvents have higher stability. They are not easily affected by other additives or by acids and bases. They have a low degree of influence on polymerization reactions. The shortest dissolution time was obtained for the polymer B3 at 2 h, and the viscosity of the solution was 34 mPa·s. The preferred cosolvent was B3.

**Table 3.** Effects of cosolvents on solubility of the polymer.

| Sample | | A1 | A2 | A3 | B1 | B2 | B3 |
|---|---|---|---|---|---|---|---|
| Concentration (mg/L) | 0 | 2000 | 2000 | 2000 | 2000 | 2000 | 2000 |
| Time (h) | 7 | 5 | 3 | 3.5 | 3 | 4 | 2 |
| Viscosity (mPa·s) | 43 | 40 | 36 | 35 | 37 | 41 | 34 |

Table 4 lists the effects of different amounts of B3 on the solubility of the polymer. When the amount of cosolvent B3 added was 2000 mg/L, the dissolution time decreased significantly, but the viscosity of the polymer decreased slightly. The amount of cosolvent was selected as 2000 mg/L of B3.

**Table 4.** Effects of different amounts of B3 on the solubility of the polymer.

| Concentration (mg/L) | 0 | 200 | 500 | 1000 | 2000 | 5000 |
|---|---|---|---|---|---|---|
| Time (h) | 7 | 6.5 | 6 | 5 | 2 | 1.2 |
| Viscosity (mPa·s) | 46 | 42 | 38 | 37 | 34 | 25 |

### 3.1.4. Analysis of Structural Characterization

Figure 8 shows the infrared spectrum of the polymer.

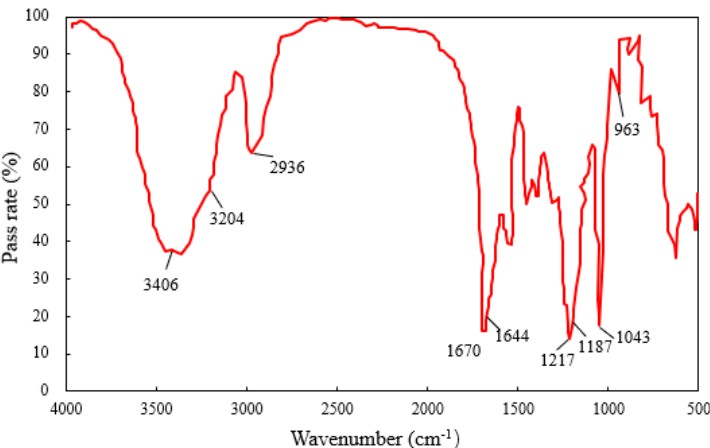

**Figure 8.** FI-IR spectrum of polymer.

In order to verify whether the polymeric monomer participated in the copolymerization, the infrared spectrum of the sample was analyzed with infrared spectroscopy. The infrared spectrum of the sample is shown in Figure 8. There were characteristic absorption peaks of sulfonate at 1217, 1187 and 1043 cm$^{-1}$, which proved the existence of AMPS; a characteristic absorption peak of pyrrolidone (-CON) at 1670 cm$^{-1}$, which proved the

existence of NVP; and a characteristic absorption peak of amide (-NH$_2$) at 3406 cm$^{-1}$, which proved the existence of AM. There was a weak characteristic absorption peak of (-COO$^-$) at 1644 cm$^{-1}$ and an absorption peak of quaternary ammonium at 963 cm$^{-1}$, which proved the existence of hydrophobic monomer DS-16. From the above absorption peaks, it can be concluded that the polymer was a copolymer of AM/AMPS/NVP/DS-16.

### 3.2. Analysis of Polymer

#### 3.2.1. Evaluation of Physical and Chemical Properties

The physical and chemical properties of the polymer sample QJ75-39 were evaluated, including the solid content, molecular weight, filtration factor, residual single content, degree of hydrolysis, insolubility and other physical and chemical properties, as presented in Table 5.

**Table 5.** Summary of physical and chemical properties.

| Sample | QJ75-39 |
| --- | --- |
| Solid content (%) | 90.14 |
| Molecular weight ($\times 10^6$) | 6.43 |
| Intrinsic viscosity (mL/g) | 1163 |
| Filter ratio | 1 |
| Residual monomer (%) | 0.01 |
| Degree of hydrolysis (%) | 5.2 |
| Insoluble fraction (%) | 0.008 |

#### 3.2.2. Influencing Factors of Polymer

In order to study whether the QJ75-39 sample was suitable for the Lu block A reservoir conditions, the influences of temperature, salinity and aging time on the 2000 mg/L polymer solution were investigated. In the experiment, in order to reduce the error, measurements were taken three times at each point and the average value was taken. Figure 9 shows the viscosity curves of the polymer under different influencing factors.

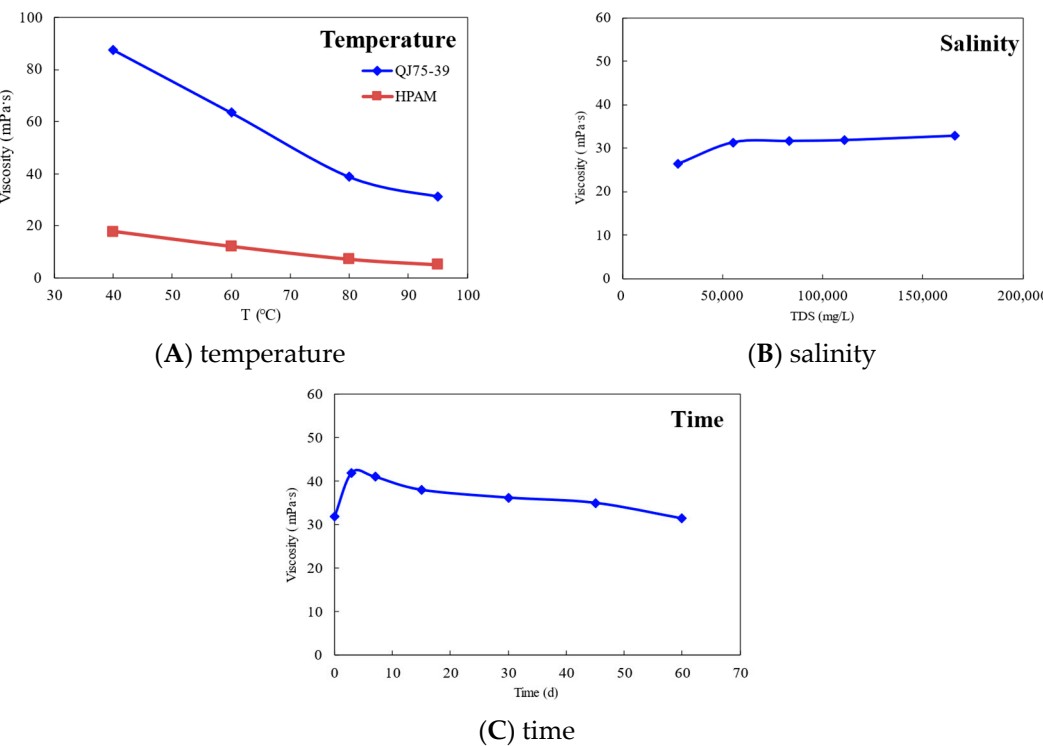

**(A)** temperature　　　　　　　　　　　　**(B)** salinity

**(C)** time

**Figure 9.** Influencing factors for the polymer.

Effect of Temperature

Figure 9A shows the viscosity curve of the polymer 2000 mg/L solution at 40~95 °C. As shown in Figure 9A, the viscosity of the polymer solution decreased with the increase in the temperature. The synthesized polymer QJ75-39 had better temperature resistance than the conventional polymer HPAM. When the temperature reached 95 °C, the viscosity of the synthesized polymer QJ75-39 was 31.3 mPa·s, while that of the conventional polymer HPAM was 5.1 mPa·s. The viscosity of QJ75-39 was much higher than HPAM at 95 °C. This was mainly due to the fact that the movement of the HPAM molecular chain accelerated with the increase in the temperature, and the spatial network structure of HPAM was destroyed to some extent. The higher the temperature was, the lower the viscosity. The synthesized polymer had better temperature resistance than conventional polymers. When the temperature reached 95 °C, the viscosity of the synthesized polymers was 31.3 mPa·s, which was mainly a result of the introduction of DS-16 hydrophobic monomers and AMPS monomers. DS-16 and AMPS have strong hydrophilic groups, so they can effectively inhibit the viscosity loss caused by the hydrogen bond breakage of amide and carboxyl groups.

Effect of Salinity

Figure 9B shows the salinity tolerance of the polymer at 95 °C. It can be seen from the experimental results that the polymer had good salinity tolerance under different salinity conditions. When the TDS was 27,688.4 mg/L, the viscosity of the polymer was only 26.3 mPa·s. When the TDS was 166,130.4 mg/L, the viscosity of the polymer was 32.8 mPa·s. The viscosity of the polymer increased slowly with the increase in salinity. This was because the introduction of salinity into the solvent shielded the electrostatic repulsion, produced dehydration, caused the molecular chain to curl up, reduced the hydrodynamic size and reduced the apparent viscosity of the common polymer solution. However, hydrophobically associating polymers form supramolecular aggregates through intermolecular association, and the supramolecular aggregates can connect with each other to form a uniform, three-dimensional network structure covering the whole system under static conditions. The formation and disassembly of the three-dimensional network structure changed reversely with the increase or decrease in the degree of hydrophobic association. The introduction of salinity also increased the polarity of the solvent, meaning that stronger association of hydrophobic groups resulted in larger hydrodynamic size and greater apparent viscosity for the polymer solution, making the polymer show good salinity tolerance.

Effect of Aging

Polymer flooding is a long-term process, so the aging stability of polymer solution and the long-term effectiveness of polymer solution viscosity are very important for polymer flooding. When the injected polymer is intended to exist in the formation for a long time, it must have good aging stability.

Figure 9C shows the viscosity changes in polymers at different aging times (95 °C, 53,376.8 mg/L). The viscosity of the polymer solution increased at first and then decreased slowly. The increase in the solution viscosity was related to the formation of intramolecular hydrogen bonds in HPAM and the arrangement of amides and carboxyl groups in HPAM molecular chains. When the molecular amides and carboxyl groups were arranged in blocks, the probability of intramolecular hydrogen bond formation was small, and the expansion of the molecular chain was mainly caused by the electrostatic repulsion between the carboxyl groups on the chain. With the increase in the number of carboxyl groups, the electrostatic repulsion increased, and the polymer chain tended to stretch, so the solution viscosity increased. At the same time, polymers rely on intermolecular association to form supramolecular aggregates, and the supramolecular aggregates can connect to each other to form a uniform three-dimensional network structure covering the whole system, so the polymer had excellent long-term stability. Under the conditions of high temperature and high salinity, the viscosity was still 31.4 mPa·s after aging for 60 days.

*3.3. Analysis of Oil Displacement Performance Evaluation*

3.3.1. Analysis of Polymer Injection Capability Experiment

EOR using polymer solution is closely related to the RF and RRF. The RF is the main parameter reflecting the mobility control and injectivity of polymer solution, and the RRF is an important parameter for the reduction of the reservoir permeability due to the retention of polymer molecules on the rock surface and porous media. At the same time, it also indirectly reflects the concentration loss in the polymer. The larger the RF is, the stronger the ability of the replacement fluid to improve the water–oil flow ratio, which is conducive to expanding the wave volume; the larger the RRF, the greater the EOR after polymer injection.

Table 6 lists the results of the polymer injection capability experiments. Figure 10 shows the variation in pressure and effluent viscosity with injection volume from the polymer injection capacity experiment. The 2000 mg/L polymer solution was injected into 300 mD and 600 mD cored, respectively. The experimental results show that the polymer could be stably injected into cores with porous media of 300~600 mD. The injection pressure remained stable after a rapid rise. The injection pressures of cores A and B were 2.5 MPa and 1.08 MPa, respectively. After rising rapidly, the viscosities of the effluents were stable at 23.7 mPa·s and 28.2 mPa·s; the viscosity retention rates were 75.7% and 90.1%. The polymer had high viscosity retention. This indicates that the polymer solution could be effectively conducted to the deeper part of the reservoir and establish certain RFs and RRFs. As shown in Table 5, the RF and RRF of core A were higher than those for core B. This indicates that the polymer solution was more likely to be retained in the low-permeability core. When the polymer was adsorbed in core pores, the effective pore volume decreased, the permeability decreased, and the RF and RRF increased. The RF and RRF could still be kept at high values (the values were 353.4 and 98.9 and 259.5 and 81.7). This shows that QJ75-39 had strong abilities to improve the fluidity ratio and to absorb and replenish under the conditions of high temperature and high salinity.

**Table 6.** Summary of results of the polymer injection capability experiment.

| No. | $K_g$ (mD) | $K_w$ (mD) | Porosity (%) | Effluent Viscosity (mPa·s) | Equilibrium Pressure (Mpa) | RF | RRF |
|-----|-----------|-----------|-------------|---------------------------|---------------------------|-------|------|
| A | 300 | 102 | 13.81 | 23.7 | 2.50 | 353.4 | 98.9 |
| B | 600 | 208 | 16.61 | 28.2 | 1.08 | 259.5 | 81.7 |

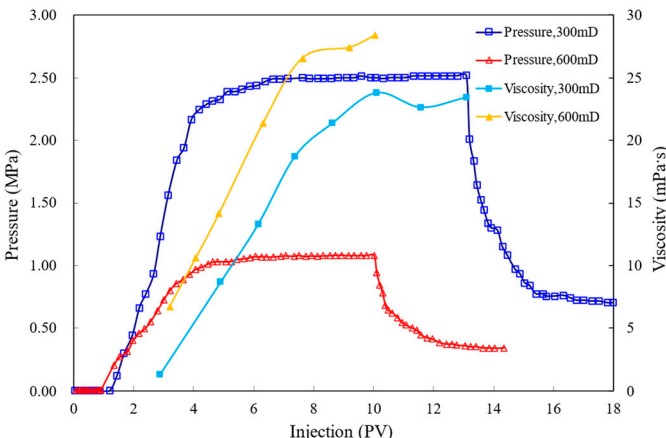

**Figure 10.** Pressure distribution diagram for polymer injection capability.

We can observe that the polymer had good injectivity and conductivity at the salinity of 53,376.8 mg/L and temperature of 95 °C at 300~600 mD.

### 3.3.2. Core Displacement Experiment Evaluation

Figure 11 shows the oil recovery, water cut and pressure difference of different polymer injection PVs.

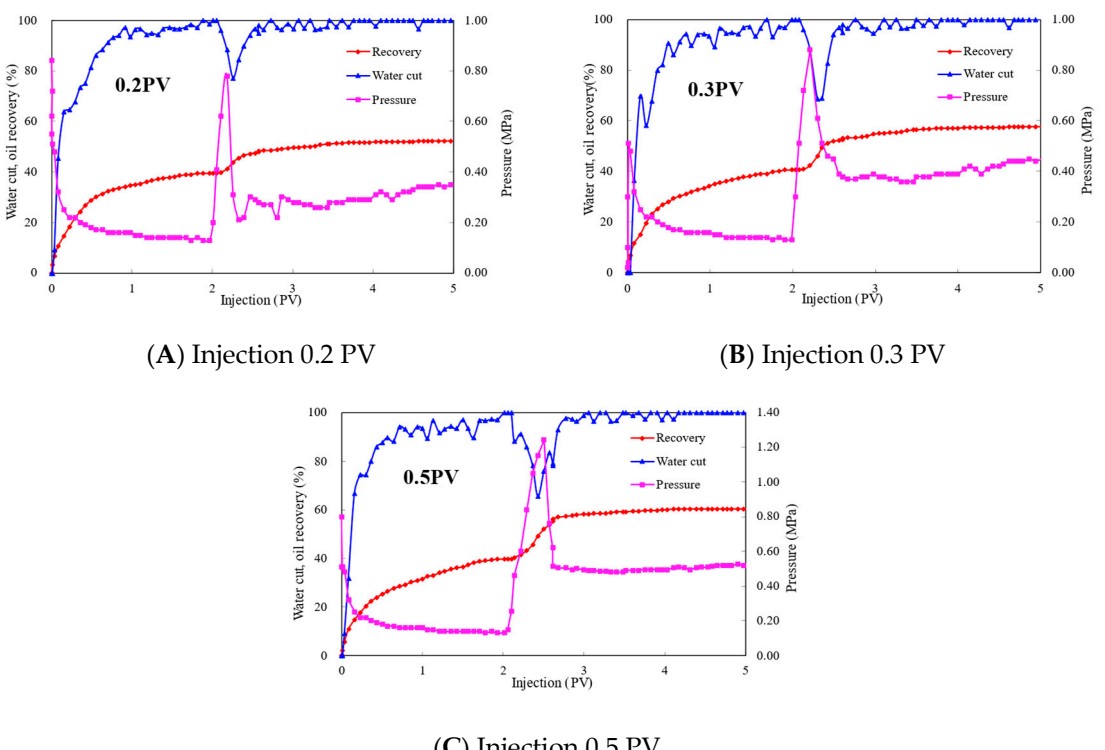

(**A**) Injection 0.2 PV

(**B**) Injection 0.3 PV

(**C**) Injection 0.5 PV

**Figure 11.** Oil displacement effect curves for different polymer injection PVs.

Table 7 shows the results of the oil displacement experiments. As shown in Figure 11, at the beginning, the injection pressure dropped rapidly and, when the oil displacement begun, the injection pressure slightly increased until it became steady. After the recovery was gradually increased to a certain value, it started to increase more slowly and, finally, remained unchanged. This was mainly because the water flooding involved a smooth migration of the oil in the formation, and the recovery levels in the water flooding stage reached 39.5%, 40.55% and 39.7%, respectively. Following the injection of the polymer, it can be seen that, when the amounts of the polymer injection were 0.2 PV, 0.3 PV and 0.5 PV, the ultimate oil recovery reached 52.26%, 57.72% and 60.35%; the lowest water cut reached 77.14%, 68.75% and 65.52%; and the EOR values were 12.76%, 17.17% and 20.65%. The pressure difference increased, and it can be seen that the polymer QJ75-39 could block the high-permeability pores and increase the sweep volume. The polymer oil recovery was higher than the water flooding, and this was due to both the improvement in the washing efficiency and the increase in the volume of the sweep.

**Table 7.** Summary of oil displacement experiment results.

| Injection (PV) | Porosity (%) | $K_w$ (mD) | So (%) | $R_w$ (%) | Ultimate Recovery (%) | EOR (%) |
| --- | --- | --- | --- | --- | --- | --- |
| 0.2 | 15.98 | 206 | 65.85 | 39.5 | 52.26 | 12.76 |
| 0.3 | 15.77 | 201.7 | 65.36 | 40.55 | 57.72 | 17.17 |
| 0.5 | 15.48 | 197.5 | 66.43 | 39.7 | 60.35 | 20.65 |

We can see that the polymer QJ75-39 could effectively enhance oil recovery, and with the increase in the injection PVs, the recovery increased and the lowest water cut decreased.

### 4. Conclusions

The main results are as follows:

(1) A new type of polymer (QJ75-39) was designed and synthesized. AMPS and NVP were introduced as temperature-resistant and salinity-tolerant monomers. DS-16 was selected as the hydrophobic monomer. $K_2S_2O_8$ + DTBP + $NaHSO_4$ was chosen as the initiator system. FI-IR showed that the polymer was a copolymer formed of AM/AMPS/NVP/DS-16;

(2) The polymer showed good temperature resistance, salinity tolerance and aging stability. When the temperature was 95 °C and the salinity was 53,376.8 mg/L, the viscosity of the polymer was 31.4 mPa·s, and the viscosity remained at 30 mPa·s after aging for 60 days. The viscosity of the polymer was 32.8 mPa·s when the salinity was 166,130.4 mg/L;

(3) According to the results of the injection capability experiments, the polymer had good injectivity and conductivity at the salinity of 53,376.8 mg/L and temperature of 95 °C at 300~600 mD. The viscosities of the effluent were 23.7 mPa·s and 28.2 mPa·s when the gas permeability values were 300 mD and 600 mD. The injection pressure could reach equilibrium quickly, and the residual resistance coefficient were 353.4 and 259.5, so it can be considered that the new polymer can be effectively transmitted to the deep part of a reservoir during polymer flooding;

(4) According to the core displacement experiment, when the amounts of the polymer injection were 0.2 PV, 0.3 PV and 0.5 PV, the EOR rates were 12.76%, 17.17% and 20.65% higher than with water flooding, which meets the performance requirements for polymer flooding under the reservoir conditions in Lu block A.

**Author Contributions:** As the first author, F.Z. was responsible for the main manuscript text, as well as for designing and conducting the experiments; Y.J. and B.W. prepared all of the figures; S.S. and D.H. critically reviewed and revised the main manuscript text; J.Z. conducted the experiments; as the corresponding author, P.L. made substantial contributions to the conception of the work, approved the final version to be published and agrees to be accountable for all aspects of the work if questions related to the accuracy or integrity of any part of the work are investigated. All authors have read and agreed to the published version of the manuscript.

**Funding:** This research was funded by the National Science Foundation of China (51774256).

**Institutional Review Board Statement:** Not applicable.

**Informed Consent Statement:** Not applicable.

**Data Availability Statement:** Not applicable.

**Acknowledgments:** The authors would like to thank the research team members for their contributions to this work.

**Conflicts of Interest:** The authors declare no conflict of interest.

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
