# Peer review of "Laboratory Experimental Study on Polymer Flooding in High-Temperature and High-Salinity Heavy Oil Reservoir"

_applsci, doi:10.3390/app122211872_

Round 1

Reviewer 1 Report

In this study, the authors designed and synthesized temperature-resistance and salinity-tolerance polymer. They used experimental results to come up with some conclusions. The manuscript may be appropriated to be published after minor revision.

Below is a list of comments which should be considered:

1.        Line37-54, there is too much introduction of reservoir. Please simplify it.

2.        The paper has some tense problems. The authors should carefully check the manuscript and modify.

3.        There is an empty blank in Table 1. Is it right?

4.        Fig.7 and Fig.9 have format errors. Please check and correct such problems in the manuscript.

5.        Some format issues of the references are observed. Please careful check and revise the manuscript. For example, the 15th reference.

6.        In conclusion, the expression is not clear enough, please summarize it again. For example, the fourth point don’t have experiment conditions.

Author Response

Dear Editors and Reviewers,

Thank you very much for your helpful comments and suggestions. According to your comments of reviewers and editor, we have carefully and meticulously revised the manuscript (applsci-2016163), and responded, point by point to the comments mentioned the page, and line numbers. The revised parts are marked in red to highlight, the amendments and deleted contents are marked with blue strike-out font to highlight deleted sections.  "Revision, changes marked" see the attachment.

Reviewer #1:

[Comment 1]:  Line37-54, there is too much introduction of reservoir. Please simplify it.

[Reply 1]: Thank you for your constructive suggestions. I have modified this part of the content according to your prompt.

[Changes]: Please see Line 55-76, in the "Revision, changes marked".

[Comment 2]: The paper has some tense problems. The authors should carefully check the manuscript and modify.

[Reply 2]: We have checked the article again and made changes to the mistakes.

[Comment 3]: There is an empty blank in Table 1. Is it right?

[Reply 3]: The empty blank represents no addition, which is the control group of this experiment.

[Comment 4]: Fig.7 and Fig.9 have format errors. Please check and correct such problems in the manuscript

[Reply 4]: According to your suggestion, I modified Figure 7 and Figure 9 in the paper to make them clearer.

[Changes]: Please see Line 55-76 on page 2, in the "Revision, changes marked".

[Comment 5]: Some format issues of the references are observed. Please careful check and revise the manuscript. For example, the 15th reference.

[Reply 5]: We have checked the article again and made changes to the mistakes.

[Comment 6]: In conclusion, the expression is not clear enough, please summarize it again. For example, the fourth point don’t have experiment conditions.

[Reply 6]: Thank you for your constructive suggestions. We have modified this part of the content according to your prompt.

[Changes]: Please see Line 432-446 on page 16, in the "Revision, changes marked".

Reviewer 2 Report

The paper titled “Laboratory Experimental Study on Polymer Flooding in High- 2 Temperature and High-salinity Heavy Oil Reservoir” can be interesting paper to the readers. In this manuscript, the authors have attempted to synthesize and design a new type of polymer same is tested for oil filed for enhanced recovery.  Unfortunately, the core samples on which these polymers were tested are artificial cores A (300 mD), and artificial core B (600 mD). How is that possible to make the core of the same oil field composition exhibit similar rock fluid and petrophysical properties? The subsurface reservoirs experience subsidence and overburden stress hence how that is possible to relate the results of artificial cores with that of the subsurface conditions. Regrettably, due to numerous flows and a complete lack of scientific rigor, I am not in the position to recommend this paper for publication. Having read the results and discussion I have reservations as it became apparent that this manuscript does not merit publication in its present state. My Major criticism is: 

(a)   The poor level of the English language is far below any international standard. The inappropriate flow of previous sentences in some places.  

(b)   Inadequate introduction with no statement of purpose explained. The reader is left with a feeling "so what" after reading your introduction. You did not specify how you designed and synthesized your polymers. You did not have validated your results.

(c)    You did not have provided field background and their maps, locations etc.

(d)   Core displacement experiment. You did not have provided the core preparation methods and photographs of how you prepared the cores and what is the length and diameters of these core samples. You have not reported anything related to experimental errors and uncertainty in measured values.

(e)    Lack of proper explanation of methodology. You did not have mentioned the field rock and fluid properties and the compatibility of the injected fluids with reservoir fluids is not reported.

(f)     Results and Discussion NEED to be individual chapters. The results are too small and they did not provide any validations. A small part of the data is presented in the manuscript. The results from published literature also do not provide any evidence to support or against it. Need explanations?

(g)   I do not understand what is the compositions of these Cosolvents: A1, A2, A3, B1, B2, B3, Beijing Chemical Plant?

(h)   From Figure 4, line number221 mention as DS-1X hydrophobic monomer belongs to the same hydrophobic monomer but has a different chain length. In this figure, I do not see anything related to DS-1 X monomer neither this type of polymer mentioned anywhere in the manuscript, please revisit this.

(i)     Conclusions section must rewrite as describing the quantitative methods are designed to provide summaries of data that support generalizations about the phenomenon under study.

Author Response

Dear Editors and Reviewers,

Thank you very much for your helpful comments and suggestions. According to your comments of reviewers and editor, we have carefully and meticulously revised the manuscript (applsci-2016163), and responded, point by point to the comments mentioned the page, and line numbers. The revised parts are marked in red to highlight, the amendments and deleted contents are marked with blue strike-out font to highlight deleted sections. "Revision, changes marked" see the attachment.

Reviewer #2:

[Comment 1]: The poor level of the English language is far below any international standard. The inappropriate flow of previous sentences in some places. 

[Reply 1]: Thank you very much for your advice. We regret there were problems with the English. The paper has been carefully revised by a native English speaker to improve the grammar and readability.

[Comment 2]: Inadequate introduction with no statement of purpose explained. The reader is left with a feeling "so what" after reading your introduction. You did not specify how you designed and synthesized your polymers. You did not have validated your results.

[Reply 2]: Thank you for your constructive suggestions. In order to make the introduction clearer. We have modified the content of the “introduction” section. And the design and synthesized have been supplemented in the “experiment” section. The design is mainly divided into three steps. 1) Matched initiations and polymerization conditions. 2) Introduced suitable temperature-resistance and salinity-tolerance monomers. 3) Introduced suitable hydrophobic monomers. FIIR verified that the synthesized polymer is the target polymer.

[Changes]: Please see Line 41-135 on page 2-4, Line 163-174 on page 5 in the "Revision, changes marked".

[Comment 3]: You did not have provided field background and their maps, locations etc.

[Reply 3]: We cannot provide the map and location of the reservoir because of confidentiality. We provided more reservoir parameters in the “introduction” section.

[Comment 4]: Core displacement experiment. You did not have provided the core preparation methods and photographs of how you prepared the cores and what is the length and diameters of these core samples. You have not reported anything related to experimental errors and uncertainty in measured values.

[Reply 4]: Thank you for your constructive suggestions. The core preparation methods is not the focus of this study. The artificial core is provided by Beijing Huarui Xincheng Technology Co., Ltd., and we provided with the parameters of the test area. Before the start of the experiment, we carried out the gas permeability test. The core picture has been supplemented to the material section. The test results are within the error range. The length and diameters of these core samples are 30 cm and 2.5 cm. We have supplemented core parameters and experimental errors in the “experiment” section.

[Changes]: Please see Line 197-199, 213-216 on page 7, in the "Revision, changes marked".

[Comment 5]: Lack of proper explanation of methodology. You did not have mentioned the field rock and fluid properties and the compatibility of the injected fluids with reservoir fluids is not reported.

[Reply 5]: Thank you very much for your suggestions. We have supplemented the properties. Through the communication with the field staff, we have supplemented the properties of rocks and fluids in the introduction. In the experiment, formation simulated water is used, and the synthetic polymer and formation simulated water have good compatibility.

[Changes]: Please see Line 120-129 on page 4, 209-211 on page 7, in the "Revision, changes marked".

[Comment 6]: Results and Discussion NEED to be individual chapters. The results are too small and they did not provide any validations. A small part of the data is presented in the manuscript. The results from published literature also do not provide any evidence to support or against it. Need explanations?

[Reply 6]: Thank you very much for your suggestions. But Results and Discussion is always a part. This paper mainly introduces a new type of polymer QJ75-39, and we mainly consider the properties, injection ability and oil displacement of the polymer. More discussions have been included in the revised manuscript. We agree that the data is small in the manuscript. But considering the time, we cannot complete more experiments. We have supplemented the temperature-resistance experiment of the conventional HPAM. By comparison, we could clearly know that QJ75-39 has better temperature resistance. The conventional HPAM is not suitable for use in high temperature and high salinity reservoirs. More research is our future research direction and will be the subject of our next manuscript.

[Changes]: Please see Line 330-334 on page 12-13, 341-345 on page 13, 375-394 on page 14, Line 416-417 on page 15 in the "Revision, changes marked".

[Comment 7]: I do not understand what is the compositions of these Cosolvents: A1, A2, A3, B1, B2, B3, Beijing Chemical Plant?

[Reply 7]: Cosolvents: A1, A2, A3, B1, B2, B3 are all non-ionic cosolvents. Their main component is fatty alcohol polyoxyethylene ether (AEO). A1, A2, A3 is the straight-chain AEO, B1, B2, B3 is the isomeric-AEO. They have different HLB. The main purpose of adding cosolvent is to improve the solubility of polymers. Nonionic cosolvents is not ionized in aqueous solution and is not easily affected by other additives, acids and bases, so it has little impact on polymerization. We added the ingredient of cosolvents in the “Materials” section.

[Changes]: Please see Line 148-150 on page 5 in the "Revision, changes marked".

[Comment 8]: From Figure 4, line number 221 mention as DS-1X hydrophobic monomer belongs to the same hydrophobic monomer but has a different chain length. In this figure, I do not see anything related to DS-1 X monomer neither this type of polymer mentioned anywhere in the manuscript, please revisit this.

[Reply 8]: DX-1X is a diallyl type double hydrophobic monomer. We have supplemented the molecular structure and related contents of DX-1X in 3.1.1. DX-1X has good water solubility and the length of hydrophobic carbon chain can be adjusted; Secondly, after polymerization, the main chain is looped, which can improve the rigidity and shear resistance of the main chain; Then, the relative activity is high, and it is easy to copolymerize with acrylamide; Finally, it has strong temperature resistance, salt resistance, thermal stability and strong intermolecular association.

[Changes]: Please see Line 253-269 on page 9 in the "Revision, changes marked".

[Comment 9]: Conclusions section must rewrite as describing the quantitative methods are designed to provide summaries of data that support generalizations about the phenomenon under study.

[Reply 9]: Thank you for your constructive suggestions. We have modified this part of the content according to your prompt. In order to make the conclusion clearer, we rewrote our conclusion.

[Changes]: Please see Line 432-446 on page 16 in the "Revision, changes marked".

Round 2

Reviewer 2 Report

Authors must have to try their best to do correction but still at some places manuscript needs corrections as mentioned below,  

1.      The length of artificial cores is not mentioned in manuscript, I am not sure whether these cores will be appropriate in size for put into the core holders. Explain?

2.      It is highly prone to errors thus the limitation of experimental set-up should be mentioned?

3.      The system consists of a core holder and an oven. So what is the use of oven in this experimental setup? Have you done experiments under changing temperature conditions?

4.      Have you performed experiments under overburden stress conditions? As the authors are saying that these sample exhibits very high permeability. Need explanations?

5.      At line number 270 it is mentioned that “Table 5. Summary of physical and chemical properties”   of what ? Properties of core or fluid or what?

6.      At line number 272 under heading “3.2.2 Influencing factors of the polymer” you must have to provide the description what and how these results were obtained and what exactly are the influencing factors under what conditions. Authors have only shown figures.

7.      Authors are advised to go through the whole manuscript again and carefully read and improve it for further consideration as advised. 

Author Response

Dear Editors and Reviewers,

Thank you very much for your helpful comments and suggestions. According to your comments of reviewers and editor, we have carefully and meticulously revised the manuscript (applsci-2016163), and responded, point by point to the comments mentioned the page, and line numbers. The revised parts are marked in red to highlight, the amendments and deleted contents are marked with blue strike-out font to highlight deleted sections. "Revision, changes marked" see the attachment.

Reviewer #2:

[Comment 1]: The length of artificial cores is not mentioned in manuscript, I am not sure whether these cores will be appropriate in size for put into the core holders. Explain?

[Reply 1]: Thank you very much for your advice. The length of artificial cores is 30 cm. These cores are appropriate in size for put into the core holders. We reorganized the content to make the “material” section clearer.

[Changes]: Please see Line 162-163 on page 6 in the "Revision, changes marked".

[Comment 2]: It is highly prone to errors thus the limitation of experimental set-up should be mentioned?

[Reply 2]: Thank you for your constructive suggestions. We have added the relevant content in the experimental section. In the process of experimental operation, we strived to be accurate in order to reduce errors. However, because of the long experimental period, it is difficult for us to carry out a large number of comparative experiments. And homogeneous cores are difficult to reflect the actual formation conditions. In next research, we will further improve the experiment and further strengthen the connectivity between physical experiments and actual situation.

[Changes]: Please see Line 128-129 on page 4, Line 140,144-145 on page 5,Line 192-194 on page 7 in the "Revision, changes marked".

[Comment 3]: The system consists of a core holder and an oven. So what is the use of oven in this experimental setup? Have you done experiments under changing temperature conditions?

[Reply 3]: Thank you very much for your advice. The oven is used to keep the temperature constant during the experiment. The experiment needs to be kept at a constant temperature. The experimental temperature is the formation temperature. In the future, we will consider the temperature change when conducting thermal recovery experiments.

[Changes]: Please see Line 169-170 on page 6 in the "Revision, changes marked".

[Comment 4]: Have you performed experiments under overburden stress conditions? As the authors are saying that these sample exhibits very high permeability. Need explanations?

[Reply 4]: Thank you for your constructive suggestions. Because of the limitations of the experimental conditions, we did not consider overburden stress conditions. In the experiment, we simulated the formation conditions by the equipment. And our experimental conditions cannot reach the formation fracture pressure. It cannot cause core cracks and caprock rupture. The permeability of the core is 300 and 600mD, which is consistent with the geological conditions. The permeability of core is not the focus of this study. And the polymer has good injectivity in the core.

[Comment 5]: At line number 270 it is mentioned that “Table 5. Summary of physical and chemical properties” of what? Properties of core or fluid or what?

[Reply 5]: Thank you very much for your suggestions. Table 5 shows the physical and chemical properties of the polymer QJ75-39. We evaluated it through the methods commonly used in the industry.

[Changes]: Please see Line 269 on page 10, in the “Revision, changes marked”.

[Comment 6]: At line number 272 under heading “3.2.2 Influencing factors of the polymer” you must have to provide the description what and how these results were obtained and what exactly are the influencing factors under what conditions. Authors have only shown figures.

[Reply 6]: Thank you very much for your suggestions. We have adjusted the “3.2.2” section of the article according to your comments and suggestions.

[Changes]: Please see Line 275-277, 289 on page 10-11, Line 314-315 on page 12, Line 356-370 on page 14 in the "Revision, changes marked".

[Comment 7]: Authors are advised to go through the whole manuscript again and carefully read and improve it for further consideration as advised.

[Reply 7]: Thank you very much for your suggestions. We have checked the article again and made changes to the mistakes.
